# Effects of Ten Different Exercise Interventions on Motor Function in Parkinson’s Disease Patients—A Network Meta-Analysis of Randomized Controlled Trials

**DOI:** 10.3390/brainsci12060698

**Published:** 2022-05-27

**Authors:** Zikang Hao, Xiaodan Zhang, Ping Chen

**Affiliations:** Department of Physical Education, Laoshan Campus, Ocean University of China, 238 Song Ling Rd., Qingdao 266100, China; haozikang@stu.ouc.edu.cn (Z.H.); zhangxiaodan@stu.ouc.edu.cn (X.Z.)

**Keywords:** exercise interventions, dance, Parkinson’s disease, network meta-analysis

## Abstract

Objective: The aim of this study was to evaluate ten exercise interventions (YOGA: yoga training, RT: resistance training, AQU: aquatic training, TAI: Taiji Qigong training, TRD: treadmill training, VR: virtual reality training, DANCE: musical dance training, WKT: walking training, CYC: cycling training, BDJ: Baduanjin Qigong training) on motor function in Parkinson’s disease (PD) patients. Design: Through searching PubMed, Embase, Cochrane Library, Web of Science, and CNKI, only randomized controlled trials (RCTs) were collected to study the effects of the ten exercise interventions on motor function in patients with Parkinson’s disease. The included studies were evaluated for methodological quality by the Cochrane bias risk assessment tool. Results: The RCTs were collected between the earliest available date and April 2022. Sixty RCTs were included and the total sample size used in the study was 2859. The results of the network meta-analysis showed that DANCE can significantly improve patients’ Berg Balance Scale (BBS) (SUCRA = 78.4%); DANCE can significantly decline patients’ Unified Parkinson’s Disease Rating Scale score (UPDRS) (SUCRA = 72.3%) and YOGA can significantly decline patients’ Timed-Up-and-Go score (TUGT) (SUCRA = 78.0%). Conclusion: Based on the network meta-analysis and SUCRA ranking, we can state that dance, yoga, virtual reality training and resistance training offers better advantages than other exercise interventions for patients’ motor function.

## 1. Introduction

Parkinson’s disease has become the second most prevalent neurodegenerative disease worldwide [1], affecting the quality of life and physical and mental health of more than six million people [2]. Parkinson’s disease can cause a number of motor dysfunctions that can seriously affect the lives of patients and place a significant burden on their families [3].

There is no complete cure for Parkinson’s disease, only a way to alleviate its symptoms to some extent [4]. Medication is currently the primary option for Parkinson’s disease relief, but the side effects and development of drug resistance or the cost of medication have limited the widespread use of medication in clinical practice and has become a long-term option for patients [5]. Is there a treatment option that is less costly and has almost no side effects? Physical exercise has made good progress in the treatment of other degenerative diseases due to its great ease of handling and the absence of side effects [6,7,8]. As a result of research and studies, it has been noted in relevant Parkinson’s disease rehabilitation studies that physical exercise can be of considerable help in improving motor function and slowing down the progression of Parkinson’s disease in people with it [9]. Previous studies have consistently shown that physical activity has considerable benefits for maintaining brain health, improving motor performance and enhancing quality of life in people with Parkinson’s disease [10].

However, for physical activity, different exercise programs have different characteristics and may have different effects on people with Parkinson’s disease, and a previous meta-analysis has only compared the effects of a particular exercise type relative to a control group for people with Parkinson’s disease [11,12,13,14]. There is still a lack of evidence-based recommendations as to which exercise programme is most suitable for improving motor function in people with Parkinson’s disease. It is therefore particularly important to find an exercise modality within a complex exercise programme that is suitable for improving the symptoms associated with motor function in patients with Parkinson’s disease, especially when physicians are considering the use of exercise prescriptions to treat patients with Parkinson’s disease.

Network meta-analysis is a recent evidence-based technique that uses direct or indirect comparisons to compare the effects of multiple interventions on a disease and to estimate the rank order of each treatment [15]. Therefore, in this study we used network meta-analysis to compare different exercise programmes (aquatic training, cycling, walking exercises, treadmill exercises, yoga exercises, taijiquan qigong, baduanjin qigong, musical dance exercises, virtual reality exercises and resistance exercises) in order to assess the effect of these exercise programmes on the motor function of Parkinson’s patients and to provide patients and clinicians with a better understanding of the effects of these programmes. The aim is to evaluate the effects of these exercise programmes on motor function in Parkinson’s patients and to provide evidence-based recommendations for patients and clinicians.

## 2. Materials and Methods

### 2.1. Search Strategy

The researchers in this paper searched five electronic databases (Pubmed, EMBASE, the Cochrane Central Register of Controlled Trials, Web of Science and CNKI) from their creation to April 2022. The search strategy was constructed around the PICOS tool: (P) Population: people with Parkinson’s disease; (I) Intervention: exercise; (C) Comparator: control group with only usual care and appropriate rehabilitation measures (including usual balance training); (O) Outcomes: motor tests for people with Parkinson’s disease. (S) Study type: RCTs. The detailed search strategy is shown in Table 1 (Pubmed is used as an example)

### 2.2. Inclusion Criteria

(1) An experimental group with different exercise modalities as an intervention for Parkinson’s disease; (2) a control group with routine care and rehabilitation of patients only; (3) a clinical randomised controlled trial; and (4) outcome indicators including at least one of the following: Unified Parkinson’s Disease Rating Scale (UPDRS) score [UPDRS or Movement Disorder Society-Unified Parkinson’s disease rating scale scores (MDS-UPDRS)], Berg Balance Scale (BBS) score, Timed-Up-and-Go (TUG) score.

### 2.3. Exclusion Criteria

(1) Studies with incomplete or unreported data; and (2) Studies from non-randomized controlled trials [including quasi-randomized controlled trials, animal studies, protocols, conference abstracts, case reports or correspondence].

### 2.4. Study Selection

The literature was screened and excluded using the literature management software Zotero. Two researchers first screened the titles of the literature for duplication, non-randomised controlled trial studies, review papers, conference papers, protocols and correspondence. The abstracts of the literature were then read by two researchers to identify literature for inclusion and to exclude literature. Finally, the remaining literature was read in full by both researchers and further identified for inclusion. During this process, both researchers independently screened the literature and finally compared the remaining literature; if it was the same then it was ultimately included, and if it was different then it was discussed and resolved by a third researcher.

### 2.5. Data Extraction

A seven-item, standardised and pre-selected data extraction form was used to record data for inclusion in the study under the following headings: (1) author, (2) year of publication, (3) country, (4) study period, (5) sample size, (6) mean age, and (7) details of the exercise intervention.

### 2.6. Risk of Bias of Individual Studies

Two researchers independently assessed the risk of bias (ROB), in accordance with the Cochrane Handbook version 5.1.0 tool for assessing ROB in RCTs. The following seven domains were considered: (1) randomized sequence generation, (2) treatment allocation concealment, blinding of (3) participants and (4) personnel, (5) incomplete outcome data, (6) selective reporting, and (7) other sources of bias. Trials were categorized into three levels of ROB by the number of components for which high ROB potentially existed: high risk (five or more), moderate risk (three or four), and low risk (two or less) [16].

### 2.7. Data Analysis

In studies where exercise is the intervention, all variables are continuous variables and are expressed as means with standard deviation (SD) [17]. Continuous variables in the study will be reported as mean difference (MD = absolute difference between the means of two groups, defined as the difference in means between the treatment and control groups and calculated using the same scale) or standardised mean difference (SMD = mean difference in outcome between groups/standard deviation of outcome between subjects, used to combine data when trials have different scales) with 95% confidence intervals (CI) and analysis. As there are certainly potential differences across studies, we chose a random effects model for analysis rather than a fixed effects model [18].

We used Stata software (version 15.1) and performed NMA aggregation and analysis using Markov chain Monte Carlo simulation chains in a Bayesian-based framework according to the PRISMA NMA instruction manual [19,20]. We used the nodal method to quantify and demonstrate the agreement between indirect and direct comparisons, calculated through the instructions in the Stata software, and if the *p*-value > 0.05, the consistency was verified [21].

Stata software was used to present and describe network diagrams of different movement interventions. In the generated network diagrams, each node represents a different motor intervention and a different control condition, and the lines connecting the nodes represent direct head-to-head comparisons between interventions. The size of each node and the width of the connecting lines are proportional to the number of studies [22].

The intervention hierarchy was summarized and reported as a P score. The P score is considered as a frequentist analogue to surface under the cumulative ranking curve (SUCRA) values and measures the extent of certainty that a treatment is better than another treatment, averaged over all competing treatments. The P score ranges from 0 to 1, where 1 indicates the best treatment with no uncertainty and 0 indicates the worst treatment with no uncertainty. While the P score or SUCRA can be usefully re-expressed as the percentage of effectiveness or acceptability of the exercise interventions, such scores should be interpreted cautiously unless there are actual clinically meaningful differences between interventions [23]. To check for the presence of bias due to small-scale studies, which may lead to publication bias in NMA, a network funnel plot was generated and visually inspected using the criterion of symmetry [24].

## 3. Results

### 3.1. Study and Identification and Selection

A total of 6431 documents were retrieved from the electronic database, and an additional nine documents were manually searched. After eliminating duplicates, the remaining 5483 documents were read for titles and abstracts, and 5176 documents were again excluded. The remaining 307 documents were read in full and 247 documents were again excluded (for reasons including non-randomised controlled trials, incomplete data, conference papers and failure to meet the interventions included in this review), leaving a final remaining 60 documents to be included in this study (Figure 1).

### 3.2. Quality Assessment of the Included Studies

Nineteen studies were defined as low risk, 11 as high risk and 30 as medium risk. Only four of these studies achieved the simultaneous blinding of subjects and measurers, but as the intervention in these studies was exercise, it was difficult to achieve simultaneous blinding of subjects and measurers as both the patients themselves and their relatives had to sign an informed consent form before the experiment was conducted. Specific details will be presented in Appendix A.

### 3.3. Characteristics of the Included Studies

In total, we included studies from 60 randomised controlled trials, which included 2859 patients diagnosed with Parkinson’s disease. Interventions in the control group included Baduanjin Qigong training (three studies) [25,26,27], walking training (three studies) [28,29,30], treadmill training (three studies) [31,32,33], aquatic training (nine studies) [34,35,36,37,38,39,40,41,42], Taiji Qigong training (eight studies) [43,44,45,46,47,48,49,50], musical dance training (10 studies) [51,52,53,54,55,56,57,58,59,60], yoga training (six studies) [61,62,63,64,65,66], cycling training (five studies) [67,68,69,70,71], resistance training (six studies) [64,72,73,74,75,76], and virtual reality training (seven studies) [57,71,77,78,79,80,81]. Thirty-one studies reported BBS as an outcome indicator, 41 studies reported UPDRS as an outcome indicator and 38 studies reported TUGT as an outcome indicator. These studies are from East Asia (17 studies), the Americas (21 studies), Europe (17 studies), Oceania (three studies) and Central Asia (two studies). The characteristics of the included studies are shown in Table 2.

### 3.4. Network Meta-Analysis

The full NMA figure will be shown in Figure 2A, Figure 3A and Figure 4A.

#### 3.4.1. Unified Parkinson’s Disease Rating Scale-Motor (UPDRS-Motor)

All *p*-values for indirect and direct comparisons between all studies were tested for consistency and inconsistency, and all *p*-values were greater than 0.05, indicating that the effect of consistency between studies was acceptable. Details will be shown in Appendix A.

The results of the network meta-analysis showed that relative to the control group’s routine measures, musical dance exercises [MD = −4.9, 95% CI = (−7.57, −2.23)], walking exercises [MD = −4.79, 95% CI = (−9.05, −0.53)], yoga exercises [MD =−4.51, 95% CI = (−8.02, −1.00)], Taijiquan practice [MD = −4.26, 95% CI = (−6.63, −1.88)], virtual reality practice [MD = −4.12, 95% CI = (−7.34, −0.91)], cycling practice [MD = −3.70, 95% CI = (−6.65, −0.75)], aquatic exercise [MD = −2.93, CL = (−5.38, −0.48)], water exercise [MD = −2.93, 95% CI = (− 5.38, −0.48)] were superior to the control group in reducing UPDRS scores, the details of which are shown in Table 3. The probability ranking of the different exercise interventions in terms of reducing UPDRS scores was ranked first in the SUCRA for dance exercises (SUCRA = 72.3% as shown in Figure 2B). 

#### 3.4.2. Timed-Up-and-Go Test (TUGT)

All *p*-values for indirect and direct comparisons between all studies were tested for consistency and inconsistency, and all *p*-values were greater than 0.05, indicating that the effect of consistency between studies was acceptable. Details are shown in Appendix A.

The results of the network meta-analysis showed that, relative to the control group (usual care) for routine measures, yoga exercises [MD = −2.4, 95% CI = (−4.14, −0.65)], resistance training [MD = −2.19, CL = (−3.41, −0.97)], water exercise exercises [MD = −1.67, 95% CI = (−3.3, −0.03)], tai chi exercise [MD = −1.56, 95% CI = (−2.59, −0.54)] and musical dance exercise [MD = −1.24. 95% CI = (−2.48, −0.01)] were superior to the control group (usual care) in reducing TUGT time; relative to the control group (balance exercise), yoga exercise [MD = −2.5, 95% CI = (−4.96, −0.04)], resistance exercises [MD = −2.3, 95% CI = (−4.18, −0.42)] and aquatic exercises [MD = −1.78, 95% CI = (−3.14, −0.42)] were better than the control group (balance exercises) in reducing TUGT time, the details of which are shown in Table 4. The probability ranking of the different exercise interventions in terms of time to TUGT reduction was ranked first by yoga practice in the SUCRA (SUCRA = 78.0% as shown in Figure 3B). 

#### 3.4.3. Berge Balance Scale

All *p*-values for indirect and direct comparisons between all studies were tested for consistency and inconsistency, and all *p*-values were greater than 0.05, indicating that the effect of consistency between studies was acceptable. Details are shown in Appendix A.

The results of the network meta-analysis showed that musical dance exercises [MD = 7.07, CL = (1.47, 12.68)] and Badaunjin Qigong exercises [MD = 5.51, 95% CI = (0.46, 10.55) were superior to the control group (balance exercises) in increasing BBS scores relative to the control group (balance exercises) for the usual measures. Relative to the control group (usual care), musical dance exercises [MD = 5.81, 95% CI = (2.45, 9.17)] and virtual reality exercises [MD = 3.6, 95% CI = (0.96, 6.24)] were superior to the control group (usual care) in terms of increasing BBS scores, the details will be shown in Table 5. The probability ranking of the different exercise interventions in terms of increasing BBS scores was ranked first by dance exercises in the SUCRA (SUCRA = 78.4% as shown in Figure 4B). 

### 3.5. Publication Bias Test

We constructed separate funnel plots for all outcome indicators to test for possible publication bias. Visual inspection of the funnel plots did not reveal any significant publication bias [82]. Details as shown in Figure 5.

## 4. Discussion

In this study we compared the effectiveness of different exercise interventions to improve motor function in people with Parkinson’s disease. A total of 60 studies including 10 different exercise programmes were included, including 2589 patients diagnosed with Parkinson’s disease, which is a fairly large sample size. Our study showed that dance practice to music was the best exercise intervention in terms of increasing BBS test scores, dance practice to music was also the best exercise intervention in terms of decreasing UPDRS-Motor scores, but yoga training showed better results in terms of decreasing the duration of TUGT. Overall, however, we believe that dance practice to music is perhaps the most appropriate intervention for improving motor function in Parkinson’s disease.

The most common symptom of Parkinson’s disease is a significant reduction in muscle control compared to the pre-existing condition, manifesting as frozen gait, reduced balance and a number of other problems. The BBS test is a comprehensive functional test that reflects the ability of Parkinson’s patients to actively shift their centre of gravity by examining their dynamic and static balance in a sitting or standing position [83], and its results are more accurate and acceptable. For Parkinson’s patients, dancing to music is a difficult and challenging exercise that requires a certain amount of proprioceptive control of the body in a state of balance before the muscle groups responsible for performing the motor function are activated [55]. During exercise, the non-periodic activities associated with dance such as starts, stops, rotations, side steps and displacements in different directions all have a beneficial effect on the training of the patient’s body responsiveness and body posture prediction, which in turn allows the basal neural network to show a larger shared network community associated with the motor cortex, resulting in an increased priority level of connectivity between the basal ganglia and the premotor areas [9,84,85], thus improving the Parkinson’s patient**.** This in turn improves the balance of Parkinson’s patients in different states and thus improves their scores on the BBS test. Our results demonstrate that dance training to music has a statistically significant beneficial effect on the balance of Parkinson’s patients compared to other exercises, and that there is a statistically significant difference compared to the control group, which is also consistent with previous studies [14,86,87].

In addition, Parkinson’s impairs motor function not only in terms of balance [88], but also in the mouth muscles associated with speech, facial muscles associated with facial expressions and limb muscles associated with alternating movements [89]. The UPDRS-III scale is the most commonly used international measure of motor function in Parkinson’s disease and provides a comprehensive measure of improvement or deterioration in motor function in Parkinson’s patients [90]. In our study, it was shown that all exercises, regardless of type, had a role in reducing UPDRS scale scores relative to the no-exercise control group, but dance training to music was the most useful exercise intervention among the different exercises included in this study, which is consistent with previous research findings [87,91,92]. As a physical activity, dance exercise not only enhances the physical and cognitive aspects of exercise, but also enhances the therapeutic effect on motor function in Parkinson’s disease with the help of rhythmic music [93]. The improvement in motor function may be the result of the increased cognitive attention of the patient while performing the exercises [94]. Imaging evidence suggests that the improvement in motor function with dance practice appears to be related to the emergence of higher-order neurological functions (for the patient, dance practice is a new form of physical activity), with altered neuroplasticity recorded by functional magnetic resonance, reflecting increased brain connectivity, particularly between the basal ganglia and cortical motor centres [95].

With age, older people experience varying degrees of decline in muscle strength and flexibility, which is common even in healthy older people [96]. The TUGT test is often used to test lower limb muscle strength in older people due to its ease of use and sensitivity [97], and our study shows that yoga and resistance training have unparalleled advantages in reducing TUGT testing times in people with Parkinson’s disease, which is in line with previous studies. The results are the same as in the original study [98]. At the same time, however, we made the further hypothesis that yoga combined with resistance training may be more beneficial than yoga or resistance training alone, with resistance training being beneficial in slowing muscle strength loss and promoting skeletal muscle hypertrophy [99] and yoga training being beneficial in maintaining joint flexibility and ligament elasticity [100], and that combining the two at the right dose (exercise duration and intensity) may have better effects. However, further studies are needed to prove our hypothesis.

Overall, our study has some clinical implications. First, dance exercises to music and yoga exercises have a significant effect in improving motor function in Parkinson’s disease. Furthermore, doctors can promote exercise as a good non-pharmacological treatment in the management of Parkinson’s disease.

## 5. Strengths and Limitations

First, our study included 60 studies and 2859 patients, which is a very large sample size, and we also built on the original review on the treatment of motor function in people with Parkinson’s disease by including two more novel interventions, aquatic exercises and virtual reality exercises, to compare with other interventions, which provides newer and more comprehensive evidence-based recommendations.

Secondly, our study shares some limitations with the studies on which it is based. Although we made every effort to control for study heterogeneity when including these original studies, heterogeneity between studies was unavoidable (e.g.**,** the proportion of studies by region and between male and female participants).

Finally, in our study, readers should interpret the results with caution because of the small number of studies and the limited head-to-head direct comparative evidence for some interventions. It also highlights the need for further expansion of the relevant studies.

## 6. Conclusions

In our study, we recommended yoga for those who wanted to improve TUG and dance for those who wanted to improve balance. Overall, however, dance exercises to the rhythm of music, yoga, virtual reality training and resistance training are the most recommended exercise prescription for Parkinson’s patients who want to improve their overall motor function.

## Figures and Tables

**Figure 1 brainsci-12-00698-f001:**
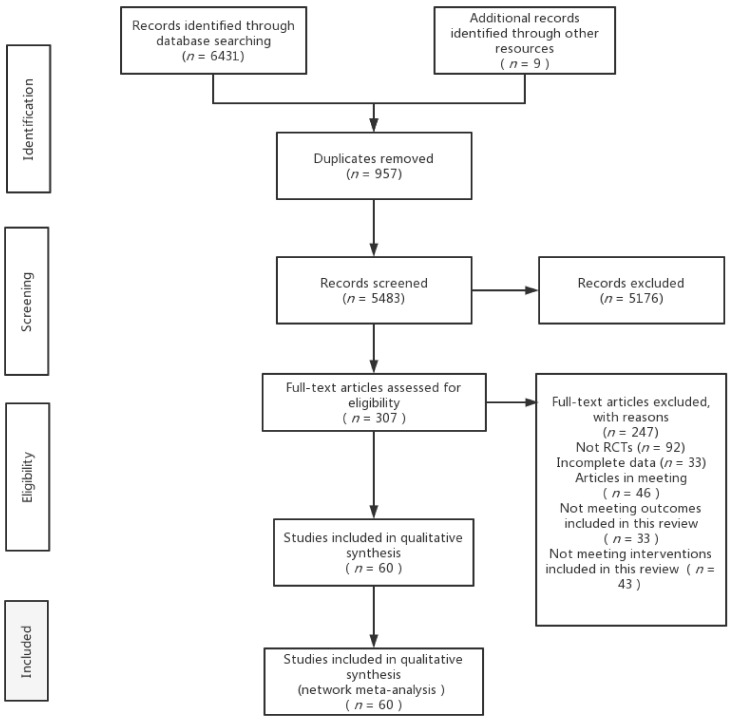
Flow diagram of literature selection.

**Figure 2 brainsci-12-00698-f002:**
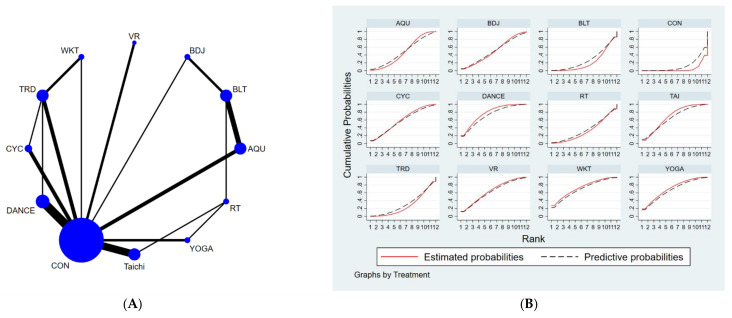
(**A**). NMA figure for UPDRS. (**B**). SUCRA plot for UPDRS.

**Figure 3 brainsci-12-00698-f003:**
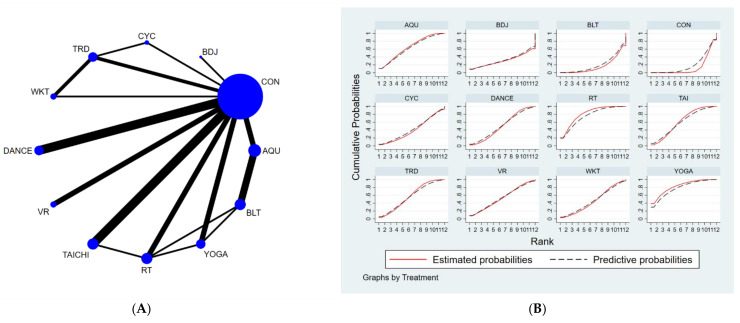
(**A**) NMA figure for TUGT, (**B**) SUCRA plot for TUGT.

**Figure 4 brainsci-12-00698-f004:**
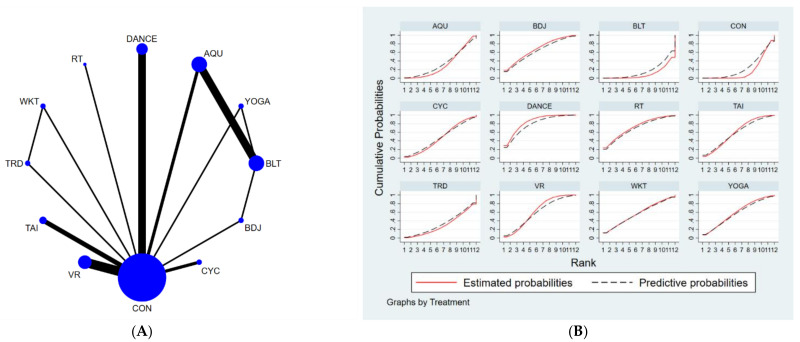
(**A**) NMA figure for BBS, (**B**) SUCRA plot for BBS.

**Figure 5 brainsci-12-00698-f005:**
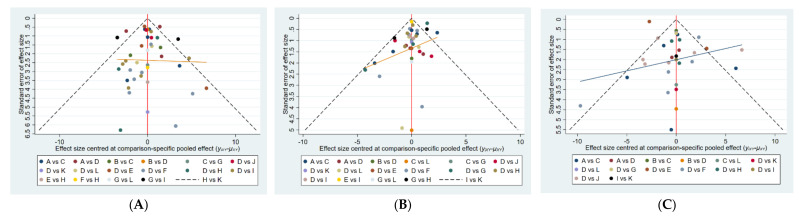
Funnel plot on publication bias. (**A**): UPDRS; (**B**): TUG; (**C**): BBS.

**Table 1 brainsci-12-00698-t001:** Search strategy on PubMed.

#1	“Parkinson disease”[MeSH]
#2	(((((Parkinson disease[Title/Abstract])OR Parkinson′s disease[Title/Abstract]) OR idiopathic Parkinson′s disease[Title/Abstract]) OR lewy body Parkinson′s disease[Title/Abstract]) OR primary Parkinsonism[Title/Abstract]) OR paralysis agitans[Title/Abstract]
#3	#1 OR #2
#4	“exercise”[MeSH]
#5	(((((((exercise[Title/Abstract]) OR exercise intervention[Title/Abstract]) OR exercise training[Title/Abstract]) OR training[Title/Abstract]) OR physical training[Title/Abstract]) OR physical exercise[Title/Abstract]) OR sports training[Title/Abstract]) OR nurse intervention[Title/Abstract]
#6	#4 OR #5
#7	randomzied controlled trials[Publication Type]
#8	#3 AND #6 AND #7

**Table 2 brainsci-12-00698-t002:** Characteristics of the studies included in the meta-analysis.

Author	Country	Year	Population	Age (Mean + SD)	Total/Male/Female	Intervention	Control	Outcome
Yuan	Taiwan	2020	Parkinson disease Hoehn & Yahr1-3	T:67.8 (5.5) C:66.5 (8.8)	T:12/2/10 C:12/9/3	VR training Length of Intervention: six weeks Freq: three times a week Duration: 30 min	CON	BBS
Pazzaglia	Italy	2020	Parkinson disease Hoehn & Yahr1-3	T:72 (7) C:70 (10)	T: 25/15/7 C: 26/17/9	VR training Length of Intervention: six weeks Freq: three times a week Duration: 40 min	CON	BBS
Xia	China	2020	Parkinson disease Hoehn & Yahr1-3	T: 65.99 (4.3) C: 66 (8.55)	T: 15/11/4 C: 15/12/3	VR training Length of Intervention: four weeks Freq: three times a week Duration: 15–20 min	CON	UPDRS, BBS
Santos	Brazil	2019	Parkinson disease Hoehn & Yahr1-3	T: 61.7 (7.3) C: 64.5 (9.8)	T: 13/11/2 C: 14/11/3	VR training Length of Intervention: eight weeks Freq: two times a week Duration: 50 min	CON	TUG, BBS
Tollar	Hungary	2019	Parkinson disease Hoehn & Yahr1-3	T: 70 (4.69) C:67.5 (4.28)	T: 25/12/13 C: 24/13/11	VR training Length of Intervention: five weeks Freq: five times a week Duration: 60 min	CON	UPDRS, BBS
Song	Australia	2018	Parkinson disease Hoehn & Yahr1-3	T: 68 (7) C: 65 (7)	T: 28/12/16 C: 25/5/20	VR training Length of Intervention: 12 weeks Freq: three times a week Duration: Minimum 15 min each time	CON	TUG
Lee	Korea	2015	NA	T: 68.4 (2.9) C: 70.1 (3.3)	T: 10/5/5 C: 10/5/5	VR training Length of Intervention: six weeks Freq: two times a week Duration: 45 min	CON	BBS
Li	China	2020	NA	T: 70.1 (3.24) C: 68.72 (3.26)	T: 40/24/16 C: 40/27/13	Dance training (Dance to the rhythm of music) Length of Intervention: four weeks Freq: seven times a week Duration: 40 min	CON	BBS
Vivas	Spain	2011	Parkinson disease Hoehn & Yahr1-3	T: 65.67 (3.67) C: 68.33 (6.92)	T: 12/3/3 C: 12/4/2	Aquatic training Length of Intervention: four weeks Freq: two times a week Duration: 45 min	BLT (balance training)	BBS
Carroll	Italy	2017	Parkinson disease Hoehn & Yahr1-3	T: 69.5 (1.75) C: 74 (6.01)	T: 10/7/3 C: 8/5/3	Aquatic training Length of Intervention: six weeks Freq: two times a week Duration: 45 min	CON	UPDRS
Volpe	Italy	2014	Parkinson disease Hoehn & Yahr1-3	T: 68 (7) C: 66 (8)	T: 17/NA/NA C: 17/NA/NA	Aquatic training Length of Intervention: eight weeks Freq: five times a week Duration: 60 min	BLT	UPDRS, BBS, TUG
Volpe	Italy	2017	Parkinson disease Hoehn & Yahr1-3	T: 70.6 (7.8) C: 70 (7.8)	T: 15/9/6 C: 15/10/5	Aquatic training Length of Intervention: eight weeks Freq: five times a week Duration: 60 min	BLT	UPDRS, BBS, TUG
Perez-de	Spain	2018	Parkinson disease Hoehn & Yahr1-3	T: 65.87 (7.09) C: 66.44 (5.73)	T: 14/NA/NA C: 15/NA/NA	Aquatic training Length of Intervention: 11 weeks Freq: two times a week Duration: 45 min	CON	UPDRS
Kurt	Turkey	2017	NA	T: 62.41 (6.76) C: 63.61 (7.18)	T: 20/11/9 C: 20/13/7	Aquatic training Length of Intervention: five weeks Freq: five times a week Duration: 60 min	BLT	UPDRS, BBS, TUG
Wang	China	2017	Parkinson disease Hoehn & Yahr1-3	T: 63.4 (7.22) C: 64.45 (6.82)	T: 20/12/8 C: 20/14/6	Aquatic training Length of Intervention: eight weeks Freq: five times a week Duration: 50 min	BLT	UPDRS, BBS, TUG
Palamara	Italy	2017	Parkinson disease Hoehn & Yahr1-2	T: 70.9 (5.7) C: 70.8 (5.3)	T: 15/9/8 C: 15/11/6	Aquatic training Length of Intervention: four weeks Freq: three times a week Duration: 60 min	CON	UPDRS, BBS, TUG
Clerici	Italy	2018	Parkinson disease Hoehn & Yahr1-2	T: 67 (8) C: 67 (11)	T: 27/NA/NA C: 25/NA/NA	Aquatic training Length of Intervention: four weeks Freq: three times a week Duration: 60 min	CON	UPDRS, BBS, TUG
Vieira	UK	2020	Parkinson disease Hoehn & Yahr1-2	T: 64.7 (1.8) C: 64.4 (3.7)	T: 25/20/5 C: 15/10/5	resistance training Length of Intervention: nine weeks Freq: two times a week Duration: 50–60 min	CON	TUG
De lima	Brazil	2019	Parkinson disease Hoehn & Yahr1-3	T: 66.2 (5.5) C: 67.2 (5.2)	T: 17/NA/NA C: 16/NA/NA	resistance training Length of Intervention: 20 weeks Freq: two times a week Duration: 30–40 min	CON	TUG
Kwok	China	2019	Parkinson disease Hoehn & Yahr1-3	T: 63.7 (8.2) C: 63.5 (9.3)	T: 71/347/34 C: 67/28/39	resistance training Length of Intervention: 20 weeks Freq: two times a week Duration: 60 min	CON	TUG, UPDRS
Leal	Brazil	2019	Parkinson disease Hoehn & Yahr1-3	T: 65.2 (2.05) C: 64.9 (2.32)	T: 27/13/14 C: 27/14/13	resistance training Length of Intervention: six MOs Freq: two times a week Duration: 20 min	CON	TUG
Schlenstedt	Germany	2015	Parkinson disease Hoehn & Yahr1-3	T: 75.7 (5.5) C: 75.7 (7.2)	T: 17/12/5 C: 15/9/6	resistance training Length of Intervention: seven weeks Freq: two times a week Duration: 60 min	CON	TUG, UPDRS
Tang	China	2019	Parkinson disease Hoehn & Yahr1-3	T: 67.76 (5.23) C: 69.64 (4.58)	T: 31/24/7 C: 31/19/12	resistance training Length of Intervention: 12 weeks Freq: two times a week Duration: 20 min	CON	BBS
Choi	Korea	2013	NA	T: 60.81 (7.6) C: 65.54 (6.8)	T: 11/NA/NA C: 9/NA/NA	Taichi training Length of Intervention: 12 weeks Freq: two times a week Duration: 60 min	CON	TUG, UPDRS
Amano	USA	2013	Parkinson disease Hoehn & Yahr1-3	T: 66 (11) C: 66 (7)	T: 15/8/7 C: 9/2/7	Taichi training Length of Intervention: 16 weeks Freq: two times a week Duration: 60 min	CON	UPDRS
Hackney	USA	2008	Parkinson disease Hoehn & Yahr1.5-3	T: 64.9 (8.3) C: 62.6 (10.2)	T: 13/2/11 C: 13/3/10	Taichi training Length of Intervention: 12 weeks Freq: two times a week Duration: 60 min	CON	UPDRS, TUG, BBS
Vergara-Diaz	USA	2018	Parkinson disease Hoehn & Yahr1-2	T: 65.7 (3.86) C: 62 (7.77)	T: 16/7/9 C: 16/9/7	Taichi training Length of Intervention: 12 weeks Freq: two times a week Duration: 60 min	CON	UPDRS, TUG
Li	USA	2012	Parkinson disease Hoehn & Yahr1-4	T: 68 (9) C: 69 (8)	T: 65/20/45 C: 65/27/38	Taichi training Length of Intervention: 24 weeks Freq: two times a week Duration: 60 min	RT	UPDRS, TUG
Choi	Korea	2016	Parkinson disease Hoehn & Yahr1-2	T: 60.81 (7.6) C: 65.54 (6.8)	T: 11/NA/NA C: 9/NA/NA	Taichi training Length of Intervention: 12 weeks Freq: one time a week Duration: 30 min	CON	TUG
Gao	Australia	2014	Parkinson disease Hoehn & Yahr1-4	T: 69.54 (7.3) C: 68.28 (8.5)	T: 37/14/23 C: 37/10/27	Taichi training Length of Intervention: 12 weeks Freq: three times a week Duration: 60 min	CON	UPDRS, TUG, BBS
You	China	2020	Parkinson disease Hoehn & Yahr1-4	T: 68.81 (5.02) C: 68.48 (5.27)	T: 35/18/17 C: 35/19/16	Taichi training Length of Intervention: eight weeks Freq: two times a week Duration: 60 min	CON	BBS, UPDRS
Xiao	China	2016	Parkinson disease Hoehn & Yahr1-4	T + C: 67.8 (9.4)	T: 49/NA/NA C: 49/NA/NA	Baduanjin training Length of Intervention: six MOs Freq: four times a week Duration: 60 min	CON	UPDRS, TUG, BBS
Shi	China	2021	Parkinson disease Hoehn & Yahr1-3	T: 67.89 (4.63) C: 67.48 (4.52)	T: 65/34/31 C: 64/34/30	Baduanjin training Length of Intervention: two MOs Freq: four times a week Duration: 30 min	BLT	BBS, UPDRS
Wang	China	2021	Parkinson disease Hoehn & Yahr1-4	T + C:65.52 (7.29)	T: 27/NA/NA C: 24/NA/NA	Baduanjin training Length of Intervention: six weeks Freq: seven times a week Duration: 30 min	BLT	UPDRS
Marieke	USA	2018	Parkinson disease Hoehn & Yahr1-3	T: 65.53 (6.1) C: 70.5 (4.4)	T: 15/5/10 C: 15/8/7	Yoga training Length of Intervention: eight weeks Freq: two times a week Duration: 60 min	CON	UPDRS
Cheung	USA	2018	Parkinson disease Hoehn & Yahr1-3	T: 63.5 (8.5) C: 65.8 (6.6)	T: 50/45/5 C: 50/45/5	Yoga training Length of Intervention: 12 weeks Freq: two times a week Duration: 60 min	CON	UPDRS
Kwok	Hong Kong	2019	Parkinson disease Hoehn & Yahr1-3	T + C: 63.6 (8.7)	T: 71/37/34 C: 67/28/39	Yoga training Length of Intervention: eight weeks Freq: one time a week Duration: 90 min	RT	UPDRS, TUG
Khuzema	India	2020	Parkinson disease Hoehn & Yahr2.5-3	T: 68.11 (4.2) C: 72 (5.2)	T: 9/NA/NA C: 9/NA/NA	Yoga training Length of Intervention: eight weeks Freq: five times a week Duration: 30–40 min	CON	TUG, BBS
Ni	USA	2016	Parkinson disease Hoehn & Yahr1-3	T + C: 72.2 (6.5)	T: 13/2/11 C: 14/5/9	Yoga training Length of Intervention: 12 weeks Freq: two times a week Duration: 60 min	CON	UPDRS, TUG, BBS
Sharma	USA	2015	Parkinson disease Hoehn & Yahr1-2	T: 62.8 (13.2) C: 73.4 (6.5)	T: 8/6/2 C: 8/3/5	Yoga training Length of Intervention: six weeks Freq: two times a week Duration: 60 min	CON	UPDRS
Song	Australia	2018	Parkinson disease Hoehn & Yahr1-3	T: 68 (7) C: 65 (7)	T: 31/16/15 C: 30/21/9	Walking training Length of Intervention: 12 weeks Freq: three times a week Duration: 15 min	CON	TUG
Michels	USA	2018	Parkinson disease Hoehn & Yahr2-2.5	T + C: 69.2 (8.7)	T: 7/NA/NA C: 6/NA/NA	Dance training (Dance to the rhythm of music) Length of Intervention: 10 weeks Freq: one time a week Duration: 60 min	CON	UPDRS, TUG, BBS
Volpe	Italy	2013	Parkinson disease Hoehn & Yahr1-2.5	T: 61.6 (4.5) C: 65.0 (5.3)	T: 12/5/7 C: 12/6/6	Dance training (Irish set dancing) Length of Intervention: 24 weeks Freq: one times a week Duration: 90 min	CON	UPDRS, BBS
Shanahan	Ireland	2017	Parkinson disease Hoehn & Yahr1-2.5	T: 69 (10) C: 69 (8)	T: 20/7/13 C: 20/7/13	Dance training (Irish set dancing) Length of Intervention: 10 weeks Freq: three times a week Duration: 20 min	CON	UPDRS
Hackney	USA	2009	Parkinson disease Hoehn & Yahr1-4	T: 68.2 (1.4) C: 66.5 (2.8)	T: 14/3/11 C: 17/5/12	Dance training (Tango) Length of Intervention: 13 weeks Freq: two times a week Duration: 60 min	CON	UPDRS, TUG, BBS
Rawson	USA	2019	Parkinson disease Hoehn & Yahr1-4	T + C: 67.2 (8.9)	T: 39/NA/NA C: 31/NA/NA	Dance training (Tango) Length of Intervention: 12 weeks Freq: two times a week Duration: 60 min	TRD	UPDRS
Duncan	USA	2012	Parkinson disease Hoehn & Yahr1-4	T: 69.3 (1.9) C: 69.0 (1.5)	T: 26/11/15 C: 26/11/15	Dance training (community-based dancing) Length of Intervention: 12 weeks Freq: two times a week Duration: 60 min	CON	UPDRS
Solla	Italy	2019	Parkinson disease Hoehn & Yahr1-3	T: 67.8 (5.9) C: 67.1 (6.3)	T: 10/4/6 C: 10/3/7	Dance training (Folk dance) Length of Intervention: 12 weeks Freq: two times a week Duration: 90 min	CON	UPDRS, TUG, BBS
Romenets	Canada	2015	Parkinson disease Hoehn & Yahr1-3	T: 63.2 (9.9) C: 64.3 (8.1)	T: 18/6/12 C: 16/9/7	Dance training (Tango) Length of Intervention: 12 weeks Freq: two times a week Duration: 60 min	CON	UPDRS, TUG
Shulman	USA	2013	Parkinson disease Hoehn & Yahr1-3	T: 66.1 (9.7) C: 65.3 (11.3)	T: 70/54/16 C: 80/62/18	Treadmill training Length of Intervention: 12 weeks Freq: three times a week Duration: 80 min	CON	UPDRS, TUG
Carvalho	Brazil	2015	Parkinson disease Hoehn & Yahr1-3	T: 64.8 (11.9) C: 64.1 (9.9)	T: 6/2/4 C: 8/2/6	Treadmill training Length of Intervention: 12 weeks Freq: two times a week Duration: 30 min	CON	UPDRS, BBS
Sage	USA	2009	Parkinson disease Hoehn & Yahr1-3	T: 65.1 (9.3) C: 64.2 (10.3)	T: 13/7/6 C: 15/11/7	Treadmill training Length of Intervention: 12 weeks Freq: three times a week Duration: 20 min	CON	UPDRS, TUG
Cugusia	Italy	2015	Parkinson disease Hoehn & Yahr1-3	T: 68.1 (8.7) C: 66.6 (7.3)	T: 10/2/8 C: 10/2/8	Walking training Length of Intervention: 12 weeks Freq: two times a week Duration: 60 min	CON	UPDRS, TUG, BBS
Bang	Korea	2017	Parkinson disease Hoehn & Yahr1-3	T: 58.3 (7.7) C: 60.6 (6.7)	T: 10/5/5 C: 10/6/4	Walking training Length of Intervention: five weeks Freq: five times a week Duration: 60 min	TRD	UPDRS, TUG, BBS
Bello	Spain	2013	Parkinson disease Hoehn & Yahr1-3	T: 58 (9.4) C: 59.5 (11.3)	T: 11/4/7 C: 11/5/6	Walking training Length of Intervention: five weeks Freq: three times a week Duration: NA	TRD	UPDRS, TUG
Kolk	Korea	2019	Parkinson disease Hoehn & Yahr1-2	T: 59.3 (8.3) C: 59.4 (9.3)	T: 65/23/42 C: 65/27/38	Cycling training Length of Intervention: 24 weeks Freq: two times a week Duration: 30 min	CON	UPDRS, BBS
Sacheli	Canada	2019	Parkinson disease Hoehn & Yahr1-3	T: 66.76 (5.9) C: 67.85 (8.5)	T: 20/7/13 C: 20/11/9	Cycling training Length of Intervention: 12 weeks Freq: three times a week Duration: 50 min	CON	UPDRS
Ridgel	USA	2019	Parkinson disease Hoehn & Yahr1-3	T: 69.9 (7.4) C: 70 (6.4)	T: 8/4/4 C: 8/3/5	Cycling training Length of Intervention: two weeks Freq: three times a week Duration: 40 min	CON	UPDRS, TUG
Arcolin	Italy	2015	Parkinson disease Hoehn & Yahr1.5-3	T: 68.7 (8.3) C: 67.8 (8.8)	T: 18/9/9 C: 13/7/6	Cycling training Length of Intervention: three weeks Freq: five times a week Duration: 60 min	TRD	UPDRS, TUG
Tollar	Hungary	2019	Parkinson disease Hoehn & Yahr2-3	T: 70.6 (4.1) C: 70 (4.69)	T: 25/13/12 C: 25/14/11	Cycling training Length of Intervention: five weeks Freq: five times a week Duration: 45 min	CON	BBS

Note: CON: control group with routine care (no exercise), BLT: control group with balance training, T: experimental group, C: control group, TRD: treadmill training. RT: resistance training, T + C: The ages of the experimental and control groups were not reported separately in the study, only the overall age was reported. UPDRS: Unified Parkinson’s Disease Rating Scale score [UPDRS or Movement Disorder Society-Unified Parkinson’s disease rating scale scores (MDS-UPDRS)], BBS: Berg Balance Scale score, TUG: Timed-Up-and-Go score, NA: unavailable, Freq: frequency.

**Table 3 brainsci-12-00698-t003:** League table on UPDRS.

DANCE	WKT	YOGA	TAI	VR	CYC	BDJ	AQU	RT	TRD	BLT	CON
DANCE	0.11 (−4.66,4.88)	0.39 (−4.02,4.80)	0.64 (−2.93,4.22)	0.78 (−3.40,4.96)	1.20 (−2.74,5.14)	1.81 (−2.48,6.10)	1.97 (−1.66,5.59)	2.91 (−1.62,7.45)	3.20 (−0.51,6.91)	3.50 (−0.49,7.49)	4.90 (2.23,7.57)
−0.11 (−4.88,4.66)	WKT	0.28 (−5.23,5.79)	0.53 (−4.34,5.40)	0.67 (−4.67,6.01)	1.09 (−3.91,6.09)	1.70 (−3.72,7.11)	1.86 (−3.04,6.76)	2.80 (−2.80,8.40)	3.09 (−0.13,6.32)	3.39 (−1.79,8.57)	4.79 (0.53,9.05)
−0.39 (−4.80,4.02)	−0.28 (−5.79,5.23)	YOGA	0.25 (−3.73,4.23)	0.39 (−4.37,5.15)	0.81 (−3.79,5.41)	1.42 (−3.33,6.17)	1.58 (−2.59,5.74)	2.52 (−1.51,6.55)	2.81 (−1.82,7.45)	3.11 (−1.28,7.51)	4.51 (1.00,8.02)
−0.64 (−4.22,2.93)	−0.53 (−5.40,4.34)	−0.25 (−4.23,3.73)	TAI	0.14 (−3.86,4.14)	0.56 (−3.23,4.35)	1.16 (−2.87,5.20)	1.33 (−2.00,4.65)	2.27 (−1.34,5.88)	2.56 (−1.30,6.42)	2.86 (−0.79,6.50)	4.26 (1.88,6.63)
−0.78 (−4.96,3.40)	−0.67 (−6.01,4.67)	−0.39 (−5.15,4.37)	−0.14 (−4.14,3.86)	VR	0.42 (−3.94,4.78)	1.03 (−3.62,5.67)	1.19 (−2.85,5.23)	2.13 (−2.74,7.01)	2.42 (−2.01,6.86)	2.72 (−1.66,7.10)	4.12 (0.91,7.34)
−1.20 (−5.14,2.74)	−1.09 (−6.09,3.91)	−0.81 (−5.41,3.79)	−0.56 (−4.35,3.23)	−0.42 (−4.78,3.94)	CYC	0.61 (−3.87,5.08)	0.77 (−3.08,4.62)	1.71 (−3.02,6.44)	2.00 (−2.01,6.01)	2.30 (−1.90,6.50)	3.70 (0.75,6.65)
−1.81 (−6.10,2.48)	−1.70 (−7.11,3.72)	−1.42 (−6.17,3.33)	−1.16 (−5.20,2.87)	−1.03 (−5.67,3.62)	−0.61 (−5.08,3.87)	BDJ	0.16 (−3.33,3.65)	1.11 (−3.57,5.78)	1.40 (−3.13,5.92)	1.69 (−1.52,4.90)	3.09 (−0.26,6.45)
−1.97 (−5.59,1.66)	−1.86 (−6.76,3.04)	−1.58 (−5.74,2.59)	−1.33 (−4.65,2.00)	−1.19 (−5.23,2.85)	−0.77 (−4.62,3.08)	−0.16 (−3.65,3.33)	AQU	0.94 (−3.15,5.04)	1.23 (−2.66,5.13)	1.53 (−0.85,3.92)	2.93 (0.48,5.38)
−2.91 (−7.45,1.62)	−2.80 (−8.40,2.80)	−2.52 (−6.55,1.51)	−2.27 (−5.88,1.34)	−2.13 (−7.01,2.74)	−1.71 (−6.44,3.02)	−1.11 (−5.78,3.57)	−0.94 (−5.04,3.15)	RT	0.29 (−4.45,5.03)	0.59 (−3.55,4.73)	1.99 (−1.68,5.65)
−3.20 (−6.91,0.51)	−3.09 (−6.32,0.13)	−2.81 (−7.45,1.82)	−2.56 (−6.42,1.30)	−2.42 (−6.86,2.01)	−2.00 (−6.01,2.01)	−1.40 (−5.92,3.13)	−1.23 (−5.13,2.66)	−0.29 (−5.03,4.45)	TRD	0.30 (−3.94,4.54)	1.70 (−1.35,4.75)
−3.50 (−7.49,0.49)	−3.39 (−8.57,1.79)	−3.11 (−7.51,1.28)	−2.86 (−6.50,0.79)	−2.72 (−7.10,1.66)	−2.30 (−6.50,1.90)	−1.69 (−4.90,1.52)	−1.53 (−3.92,0.85)	−0.59 (−4.73,3.55)	−0.30 (−4.54,3.94)	BLT	1.40 (−1.57,4.37)
**−4.90 (−7.57,−2.23)**	**−4.79 (−9.05,−0.53)**	**−4.51 (−8.02,−1.00)**	**−4.26 (−6.63,−1.88)**	**−4.12 (−7.34,−0.91)**	**−3.70 (−6.65,−0.75)**	−3.09 (−6.45,0.26)	**−2.93 (−5.38,−0.48)**	−1.99 (−5.65,1.68)	−1.70 (−4.75,1.35)	−1.40 (−4.37,1.57)	CON

YOGA: yoga training, RT: resistance training, AQU: aquatic training, TAI: Taiji Qigong training, TRD: treadmill training, VR: virtual reality training, DANCE: musical dance training, WKT: walking training, CYC: cycling training, BDJ: Baduanjin Qigong training, BLT: control group (with balance training), and CON: control group (no exercise).

**Table 4 brainsci-12-00698-t004:** League table on TUGT.

YOGA	RT	AQU	TAI	TRD	VR	DANCE	WKT	CYC	BDJ	BLT	CON
YOGA	0.21 (−1.79,2.21)	0.73 (−1.61,3.07)	0.83 (−1.16,2.83)	0.98 (−1.28,3.25)	1.09 (−1.48,3.66)	1.15 (−0.98,3.29)	1.33 (−1.17,3.83)	1.52 (−1.12,4.15)	2.30 (−2.07,6.67)	2.50 (0.04,4.96)	2.40 (0.65,4.14)
−0.21 (−2.21,1.79)	RT	0.52 (−1.32,2.37)	0.63 (−0.76,2.01)	0.77 (−1.12,2.67)	0.88 (−1.38,3.14)	0.95 (−0.77,2.67)	1.12 (−1.02,3.27)	1.31 (−0.99,3.61)	2.09 (−2.10,6.28)	2.30 (0.42,4.18)	2.19 (0.97,3.41)
−0.73 (−3.07,1.61)	−0.52 (−2.37,1.32)	AQU	0.10 (−1.78,1.99)	0.25 (−1.93,2.43)	0.36 (−2.13,2.85)	0.43 (−1.62,2.48)	0.60 (−1.83,3.03)	0.79 (−1.78,3.36)	1.57 (−2.76,5.90)	1.78 (0.42,3.14)	1.67 (0.03,3.30)
−0.83 (−2.83,1.16)	−0.63 (−2.01,0.76)	−0.10 (−1.99,1.78)	TAI	0.15 (−1.62,1.92)	0.26 (−1.89,2.40)	0.32 (−1.28,1.92)	0.50 (−1.56,2.56)	0.69 (−1.54,2.91)	1.46 (−2.67,5.60)	1.67 (−0.37,3.72)	1.56 (0.54,2.59)
−0.98 (−3.25,1.28)	−0.77 (−2.67,1.12)	−0.25 (−2.43,1.93)	−0.15 (−1.92,1.62)	TRD	0.11 (−2.27,2.48)	0.17 (−1.73,2.08)	0.35 (−1.13,1.83)	0.54 (−1.18,2.25)	1.32 (−2.94,5.58)	1.52 (−0.83,3.88)	1.42 (−0.03,2.86)
−1.09 (−3.66,1.48)	−0.88 (−3.14,1.38)	−0.36 (−2.85,2.13)	−0.26 (−2.40,1.89)	−0.11 (−2.48,2.27)	VR	0.07 (−2.19,2.33)	0.24 (−2.37,2.85)	0.43 (−2.32,3.18)	1.21 (−3.22,5.64)	1.42 (−1.25,4.08)	1.31 (−0.57,3.19)
−1.15 (−3.29,0.98)	−0.95 (−2.67,0.77)	−0.43 (−2.48,1.62)	−0.32 (−1.92,1.28)	−0.17 (−2.08,1.73)	−0.07 (−2.33,2.19)	DANCE	0.17 (−1.99,2.34)	0.36 (−1.96,2.69)	1.14 (−3.05,5.34)	1.35 (−0.86,3.56)	1.24 (0.01,2.48)
−1.33 (−3.83,1.17)	−1.12 (−3.27,1.02)	−0.60 (−3.03,1.83)	−0.50 (−2.56,1.56)	−0.35 (−1.83,1.13)	−0.24 (−2.85,2.37)	−0.17 (−2.34,1.99)	WKT	0.19 (−1.99,2.37)	0.97 (−3.42,5.36)	1.18 (−1.38,3.73)	1.07 (−0.73,2.86)
−1.52 (−4.15,1.12)	−1.31 (−3.61,0.99)	−0.79 (−3.36,1.78)	−0.69 (−2.91,1.54)	−0.54 (−2.25,1.18)	−0.43 (−3.18,2.32)	−0.36 (−2.69,1.96)	−0.19 (−2.37,1.99)	CYC	0.78 (−3.69,5.25)	0.99 (−1.70,3.68)	0.88 (−1.10,2.86)
−2.30 (−6.67,2.07)	−2.09 (−6.28,2.10)	−1.57 (−5.90,2.76)	−1.46 (−5.60,2.67)	−1.32 (−5.58,2.94)	−1.21 (−5.64,3.22)	−1.14 (−5.34,3.05)	−0.97 (−5.36,3.42)	−0.78 (−5.25,3.69)	BDJ	0.21 (−4.21,4.62)	0.10 (−3.91,4.11)
**−2.50 (−4.96,−0.04)**	**−2.30 (−4.18,−0.42)**	**−1.78 (−3.14,−0.42)**	−1.67 (−3.72,0.37)	−1.52 (−3.88,0.83)	−1.42 (−4.08,1.25)	−1.35 (−3.56,0.86)	−1.18 (−3.73,1.38)	−0.99 (−3.68,1.70)	−0.21 (−4.62,4.21)	BLT	−0.11 (−1.96,1.75)
**−2.40 (−4.14,−0.65)**	**−2.19 (−3.41,−0.97)**	**−1.67 (−3.30,−0.03)**	**−1.56 (−2.59,−0.54**)	−1.42 (−2.86,0.03)	−1.31 (−3.19,0.57)	**−1.24 (−2.48,−0.01)**	−1.07 (−2.86,0.73)	−0.88 (−2.86,1.10)	−0.10 (−4.11,3.91)	0.11 (−1.75,1.96)	CON

YOGA: yoga training, RT: resistance training, AQU: aquatic training, TAI: Taiji Qigong training, TRD: treadmill training, VR: virtual reality training, DANCE: musical dance training, WKT: walking training, CYC: cycling training, BDJ: Baduanjin Qigong training, BLT: control group (with balance training), and CON: control group (no exercise).

**Table 5 brainsci-12-00698-t005:** League table on BBS.

**DANCE**	**RT**	**BDJ**	**VR**	**TAI**	**YOGA**	**WKT**	**CYC**	**AQU**	**TRD**	**CON**	**BLT**
DANCE	−0.91 (−7.48,5.66)	−1.56 (−8.43,5.30)	−2.21 (−6.47,2.05)	−2.31 (−7.20,2.58)	−2.37 (−8.27,3.53)	−2.49 (−9.72,4.73)	−3.45 (−8.79,1.89)	−4.85 (−10.06,0.36)	−5.07 (−11.71,1.57)	−5.81 (−9.17,−2.45)	−7.07 (−12.68,−1.47)
0.91 (−5.66,7.48)	RT	−0.65 (−8.92,7.61)	−1.30 (−7.53,4.93)	−1.40 (−8.07,5.27)	−1.46 (−8.96,6.05)	−1.58 (−10.12,6.96)	−2.54 (−9.52,4.44)	−3.94 (−10.85,2.96)	−4.16 (−12.21,3.89)	−4.90 (−10.54,0.74)	−6.16 (−13.37,1.04)
1.56 (−5.30,8.43)	0.65 (−7.61,8.92)	BDJ	−0.64 (−7.22,5.94)	−0.75 (−7.75,6.26)	−0.80 (−7.91,6.30)	−0.93 (−9.72,7.86)	−1.89 (−9.21,5.43)	−3.29 (−8.89,2.31)	−3.51 (−11.83,4.82)	−4.25 (−10.28,1.79)	−5.51 (−10.55,−0.46)
2.21 (−2.05,6.47)	1.30 (−4.93,7.53)	0.64 (−5.94,7.22)	VR	−0.10 (−4.53,4.33)	−0.16 (−5.76,5.44)	−0.29 (−7.22,6.64)	−1.24 (−6.13,3.65)	−2.64 (−7.42,2.13)	−2.86 (−9.18,3.45)	−3.60 (−6.24,−0.96)	−4.87 (−10.07,0.34)
2.31 (−2.58,7.20)	1.40 (−5.27,8.07)	0.75 (−6.26,7.75)	0.10 (−4.33,4.53)	TAI	−0.06 (−6.15,6.04)	−0.18 (−7.51,7.15)	−1.14 (−6.57,4.29)	−2.54 (−7.88,2.80)	−2.76 (−9.51,3.99)	−3.50 (−7.05,0.06)	−4.76 (−10.48,0.96)
2.37 (−3.53,8.27)	1.46 (−6.05,8.96)	0.80 (−6.30,7.91)	0.16 (−5.44,5.76)	0.06 (−6.04,6.15)	YOGA	−0.13 (−8.21,7.95)	−1.08 (−7.55,5.38)	−2.48 (−8.14,3.17)	−2.70 (−10.27,4.87)	−3.44 (−8.39,1.51)	−4.71 (−10.38,0.97)
2.49 (−4.73,9.72)	1.58 (−6.96,10.12)	0.93 (−7.86,9.72)	0.29 (−6.64,7.22)	0.18 (−7.15,7.51)	0.13 (−7.95,8.21)	WKT	−0.96 (−8.58,6.66)	−2.36 (−9.90,5.19)	−2.58 (−8.21,3.06)	−3.32 (−9.73,3.09)	−4.58 (−12.40,3.24)
3.45 (−1.89,8.79)	2.54 (−4.44,9.52)	1.89 (−5.43,9.21)	1.24 (−3.65,6.13)	1.14 (−4.29,6.57)	1.08 (−5.38,7.55)	0.96 (−6.66,8.58)	CYC	−1.40 (−7.12,4.32)	−1.62 (−8.68,5.44)	−2.36 (−6.47,1.75)	−3.62 (−9.70,2.46)
4.85 (−0.36,10.06)	3.94 (−2.96,10.85)	3.29 (−2.31,8.89)	2.64 (−2.13,7.42)	2.54 (−2.80,7.88)	2.48 (−3.17,8.14)	2.36 (−5.19,9.90)	1.40 (−4.32,7.12)	AQU	−0.22 (−7.20,6.77)	−0.96 (−4.94,3.02)	−2.22 (−5.20,0.75)
5.07 (−1.57,11.71)	4.16 (−3.89,12.21)	3.51 (−4.82,11.83)	2.86 (−3.45,9.18)	2.76 (−3.99,9.51)	2.70 (−4.87,10.27)	2.58 (−3.06,8.21)	1.62 (−5.44,8.68)	0.22 (−6.77,7.20)	TRD	−0.74 (−6.48,5.00)	−2.00 (−9.29,5.28)
**5.81 (2.45,9.17)**	4.90 (−0.74,10.54)	4.25 (−1.79,10.28)	**3.60 (0.96,6.24)**	3.50 (−0.06,7.05)	3.44 (−1.51,8.39)	3.32 (−3.09,9.73)	2.36 (−1.75,6.47)	0.96 (−3.02,4.94)	0.74 (−5.00,6.48)	CON	−1.26 (−5.75,3.22)
**7.07 (1.47,12.68)**	6.16 (−1.04,13.37)	**5.51 (0.46,10.55)**	4.87 (−0.34,10.07)	4.76 (−0.96,10.48)	4.71 (−0.97,10.38)	4.58 (−3.24,12.40)	3.62 (−2.46,9.70)	2.22 (−0.75,5.20)	2.00 (−5.28,9.29)	1.26 (−3.22,5.75)	BLT

YOGA: yoga training, RT: resistance training, AQU: aquatic training, TAI: Taiji Qigong training, TRD: treadmill training, VR: virtual reality training, DANCE: musical dance training, WKT: walking training, CYC: cycling training, BDJ: Baduanjin Qigong training, BLT: control group (with balance training), and CON: control group (no exercise).

## Data Availability

The data that support the findings of the study are available from the first author, upon reasonable request.

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
