# Peer review of "Effects of Ten Different Exercise Interventions on Motor Function in Parkinson’s Disease Patients—A Network Meta-Analysis of Randomized Controlled Trials"

_brainsci, 2022, doi:10.3390/brainsci12060698_

Round 1
Reviewer 1 Report
Dear Authors,
This is s very interesting and important study that compared the effectiveness of different exercise interventions to improve motor function in people with Parkinson's disease. A total of 60 studies including 10 different exercise programmes were included, including 2589 patients diagnosed with Parkinson's disease.
I congratulate the authors on this methodologically well-conducted, clinically important, pertinent and up-to-dated work.
I only have a few minor comments to share to improve this manuscript:
In general, in my opinion, the captions of the figures are too synthetic, making their interpretation difficult for readers who are less familiar with this type of presentation of results. The Figure 4B. is incompleted “SUCRA plot for”…?
Line 21: typo “DABCE”
Results: Line 183. It is not usual to start sentences with numerals.
Author Response
亲爱的审稿人,
我们非常感谢您对本文修订的评论。您的评论突出了您对文章细节的控制以及您在审核过程中的关心,并且我们对您提到的问题进行了更改,我们希望这些更改能够得到您的批准。
祝您工作顺利,享受生活!
对审稿人 1 的回复 评论
第1点:总的来说,在我看来,这些数字的标题过于综合,使得那些不太熟悉这种结果呈现方式的读者很难解释它们。图4B.是不完整的“苏格拉图”...?
响应 1: 对排行榜的位置进行了调整,在一定程度上帮助读者理解它们。此外,“结果”部分中的图片和表格在“材料和方法”部分中进行了详细解释。
第2点:第21行:拼写错误“DABCE”
响应 2:我们对此错误表示歉意,因为我们在上传文件时做了疏忽,并且拼写错误,单词的正确拼写应该是“DANCE”,我们已经在文章中更正了这一点。
第3点:结果:第 183 行。通常以数字开头的句子并不常见。
响应 3:这些研究来自东亚(17项研究),美洲(21项研究),欧洲(17项研究),大洋洲(3项研究)和中亚(2项研究)。
Reviewer 2 Report
My compliments for a good and quality paper
Author Response
Dear reviewer,
Thank you very much for recognising our articles and work from your professional point of view and for taking your valuable time to review them. Your approval is our greatest help.
Good luck with your work and enjoy your life!
Reviewer 3 Report
Dear authors,
Thank you for the opportunity to review your manuscript. It is a work that focuses on a topic of intriguing importance, especially in light of the power of statistical intervention:
Mentioning the software in the abstract is superfluous. The presented MD refers to what? because the ranking table orders the interventions by placing the network above the diagonal.. and below the pairwise. Thus, considering the use of STATA, have you thought about conducting SUCRA?
30 Remove According to previous research on
33 put the 3 pivotal signs .. myotonia?
61 hydrokinesitherapy
150 You may miss that “High odds in P score or SUCRA mean better results” reference: http://dx.doi.org/10.1016/j.ctcp.2020.101260
I would recommend with the characteristics of the studies a list / table on the type of intervention .. because DANCE includes very varied interventions .. irish set, tango, folk and community dance ..
Figure 2C it is not intelligible... it is fundamental because almost all interventions are indirectly versus control
2B It doesn't seem to me that the area under the DANCE curve transmits 82% .. I would like to know all the SUCRA scores
313 It is not readable, but still not necessary for the type of study in my opinion
analyzing the curves, I would like you to reconsider the calculations of the SUCRAs .. because I do not think there are high probability values and above all the interventions I think are very matched, so that there are several therapies that have excellent and similar results and not just one at the top of the ranking table
So I recommend a major revision
Author Response
Dear reviewer,
We sincerely thank you for taking the time to advise us on our article as a leading expert in the field. The issues you raised were not something we had originally thought of, as we do not have the same level of control and expertise as you, so we have made the changes you suggested, which have made the article look more fluid and scientific. Your advice, as a senior member of the field, has also deepened our understanding of the field and, to a large extent, updated our understanding of statistical methods.
Response to Reviewer 3 Comments
Point 1: Mentioning the software in the abstract is superfluous. The presented MD refers to what? because the ranking table orders the interventions by placing the network above the diagonal.. and below the pairwise. Thus, considering the use of STATA, have you thought about conducting SUCRA?
Response 1:
- Following your suggestion, we have removed the description of the software from the original text.
- We will highlight in the abstract section that we are using the SUCRA ranking to determine the likely effects of different exercise interventions.
Point 2: 30 Remove According to previous research on
Response 2: We removed " Remove According to previous research on " and replaced the sentence with “Parkinson's disease has become the second most prevalent neurodegenerative disease worldwide. ”
Point 3: 33 put the 3 pivotal signs .. myotonia?
Response 3: We may have different interpretations of the term "myotonia", but in order not to create ambiguity in the article after publication, we have removed the three main signs " resting tremor, myotonia and postural gait disorders ".
Point 4: 61 hydrokinesitherapy
Response 4: This is a mistake in our writing, we have replaced "water exercise" with "aquatic training" which is often found in this article, which is the same expression as your recommendation "hydrokinesitherapy".
Point 5: 150 You may miss that “High odds in P score or SUCRA mean better results” reference: http://dx.doi.org/10.1016/j.ctcp.2020.101260
Response 5: Thank you for the reminder that we have cited the literature you recommended, which does make the text in this section look more scientific.
Point 6: I would recommend with the characteristics of the studies a list / table on the type of intervention .. because DANCE includes very varied interventions .. irish set, tango, folk and community dance ..
Response 6: Following your reminder, we have added the specific characteristics of the "dance" to the detailed descriptions of the different interventions in Table 2.
Point 7: Figure 2C it is not intelligible... it is fundamental because almost all interventions are indirectly versus control
Response 7: We have revisited these images during the revision process and have found your comments to be very professional and useful. It is true that for readers who are not familiar with network meta-analysis, it is difficult to understand these images, so we have decided, as you suggested, to remove them in order to give our readers a better reading experience.
Point 8: 2B It doesn't seem to me that the area under the DANCE curve transmits 82% .. I would like to know all the SUCRA scores
Response 8: As described in the respomse to point 10, we have reworked the calculations according to the method in the article you recommended and all SUCRA scores will be presented in Tables 1,2,3.
Point 9: 313 It is not readable, but still not necessary for the type of study in my opinion
Response 9: We may have presented this section in the article because of the different means we use for the stata software. However, due to the valuable comments you have given us as a professional reviewer, we prefer to listen to your suggestions, so we have decided to remove this section in order to make the article more readable.
Point 10: analyzing the curves, I would like you to reconsider the calculations of the SUCRAs .. because I do not think there are high probability values and above all the interventions I think are very matched, so that there are several therapies that have excellent and similar results and not just one at the top of the ranking table
Response 10:
- In terms of the interpretation of the SUCRA problem, we believe that it may be due to a discrepancy with the code in the stata software you used, which we used before, the code " sucra prob*, labels(A B C D E F) rankog".
- Due to your expertise in this field, we have re-read the references you sent us and have used the relevant codes to calculate the area of SUCRA and its corresponding graph. We have modified the recalculated value and the graph it presents in the original text. Here are three tables of all the SUCRA values you would like to know about.
- Perhaps our understanding of these exercise interventions is not as deep as yours, so we have focused too much on the one-sided top rankings, so we have rewritten the conclusions in the abstract section based on your suggestions.
Table 1 SUCRA for BBS
|
Treatm~t |
SUCRA |
PrBest |
MeanRank |
|
AQU |
34.9 |
0.9 |
8.2 |
|
BDJ |
63.2 |
15.4 |
5.0 |
|
BLT |
16.8 |
0.1 |
10.2 |
|
CON |
24.4 |
0.1 |
9.3 |
|
CYC |
47.7 |
3.8 |
6.8 |
|
DANCE |
78.4 |
25.0 |
3.4 |
|
RT |
68.9 |
20.7 |
4.4 |
|
TAI |
58.8 |
6.7 |
5.5 |
|
TRD |
33.9 |
2.0 |
8.3 |
|
VR |
59.3 |
5.4 |
5.5 |
|
WKT |
56.3 |
11.8 |
5.8 |
|
YOGA |
57.3 |
8.2 |
5.7 |
Table 2 SUCRA for UPDRS
|
Treatm~t |
SUCRA |
PrBest |
MeanRank |
|
AQU |
48.3 |
3.5 |
6.7 |
|
BDJ |
49.9 |
5.3 |
6.5 |
|
BLT |
28.4 |
0.5 |
8.9 |
|
CON |
12.3 |
0.1 |
10.6 |
|
CYC |
57.5 |
7.7 |
5.7 |
|
DANCE |
72.3 |
18.3 |
4.1 |
|
RT |
35.9 |
2.1 |
8.1 |
|
TAI |
64.8 |
10.9 |
4.9 |
|
TRD |
31.8 |
0.6 |
8.5 |
|
VR |
62.4 |
11.8 |
5.1 |
|
WKT |
69.4 |
22.5 |
4.4 |
|
YOGA |
67.0 |
16.8 |
4.6 |
Table 3 SUCRA for TUGT
|
Treatm~t |
SUCRA |
PrBest |
MeanRank |
|
AQU |
62.1 |
10.5 |
5.2 |
|
BDJ |
32.4 |
8.6 |
8.4 |
|
BLT |
20.5 |
0.3 |
9.7 |
|
CON |
19.8 |
0.1 |
9.8 |
|
CYC |
42.6 |
3.6 |
7.3 |
|
DANCE |
51.6 |
3.8 |
6.3 |
|
RT |
75.6 |
19.3 |
3.7 |
|
TAI |
60.1 |
5.8 |
5.4 |
|
TRD |
56.8 |
5.5 |
5.8 |
|
VR |
53.6 |
8.4 |
6.1 |
|
WKT |
46.9 |
4.4 |
6.8 |
|
YOGA |
78.0 |
29.8 |
3.4 |

Round 2
Reviewer 3 Report
Dear authors, the manuscript has improved substantially, especially from a methodological point of view. I suggest its suitability for publication